# Med-HVL: Automatic Medical Domain Hallucination Evaluation for Large Vision-Language Models

## Qianqi Yan, Xuehai He, Xin Eric Wang

[1]University of California, Santa Cruz

## Abstract

Advancements in Large Vision-Language Models (LVLMs) have made significant progress in integration of visual and textual data. However, their deployment in the medical domain is impeded by critical issues of hallucinations, asking for reliable evaluation metrics and methods. We define two novel metrics: Object Hallucination and Domain Knowledge Hallucination to quantify the hallucination of LVLMs in the medical domain. We propose a scalable, automated evaluation framework, Med-HVL, to assess and mitigate hallucinations at both object and domain-knowledge levels. We reveal a significant presence of hallucinations in the LVLMs, emphasizing the need for domain-specific adaptations and finetuning to enhance their reliability for medical applications.

## 1 Introduction

Medicine is an intrinsically multimodal discipline. Despite advancements in Large Vision-Language Models (LVLMs) such as GPT4V (OpenAI 2023), LLaVA (Liu et al. 2023), and MiniGPT-v2 (Chen et al. 2023a), which enables more nuanced understanding and generation of content that blends both visual and linguistic elements, their adoption in the medical sector is met with caution due to concerns about accuracy, coherence, and the risk of generating erroneous or 'hallucinated' content. Studies has shown that the latest version of GPT-4V is not yet suitable for real-world medical diagnostics due to its inconsistent accuracy (Yan et al. 2023). At the same time, models specifically designed for the biomedical field such as LLaVA-Med (Li et al. 2023), MedBLIP (Chen et al. 2023b) are emerging. The key to implementing and deploying LVLMs in healthcare is to make these models trustworthy (Ahmad, Yaramis, and Roy 2023).

The hallucination of Large Language Models (LLMs) presents a significant challenge (Bang et al. 2023), where the complexity of language and visual data results in a vast space of possible interpretations, making it difficult for models to align their outputs with ground truth consistently. This is particularly problematic when combining LLMs with visual input data. In casual use, such inaccuracies might be inconsequential, but in the medical field, they can lead to critical health risks by impacting patient care and medical decisions. The nuanced nature of medical knowledge demands an understanding that LLMs might not consistently provide, leading to the generation of plausible yet incorrect information. Thus, developing methods to assess and mitigate these hallucinations is crucial both academically and practically.

Most existing research relies on labor-intensive human annotation, making it inefficient and difficult to scale. Automatic object-level hallucination evaluation such as Caption Hallucination Assessment with Image Relevance (CHAIR) (Rohrbach et al. 2019) aims to identify hallucinated objects in captions, requiring complex, human-devised rules for exact matching. However, these methods haven't been adapted to the specific generative styles of LVLMs, particularly in the medical context, and are prone to classification errors. Our study introduces an automated LLM-based approach to apply the CHAIR metric in medical image captioning tasks. Additionally, we have developed a Domain Knowledge Hallucination (DKH) metric specifically focused on domain-level hallucinations in LVLMs.

Our contributions are twofold: First, we formally define hallucination in the medical domain, categorizing it as **Object Hallucination** and **Domain Knowledge Hallucination**; Second, we propose Med-HVL, an automatic method for evaluating hallucinations in the medical domain, which is more stable and scalable and effectively assesses hallucinations at both the object and domain knowledge levels.

## 2 Metrics

We introduce two pivotal metrics to quantify the specific types of inaccuracies or 'hallucinations' that can occur in medical image captioning.

### 2.1 Object Hallucination

CHAIR is a widely recognized metric for assessing object hallucination in image captioning tasks. Given the ground truth objects in the image, CHAIR calculates the proportion of objects that appear in the caption but not the image (Rohrbach et al. 2019). One of its variants evaluates the hallucination degree at the object instance level and can be formulated as follows:

$$\text{CHAIR}_I = \frac{|\{\text{ hallucinated objects }\}|}{|\{\text{ all mentioned objects }\}|} \quad (1)$$

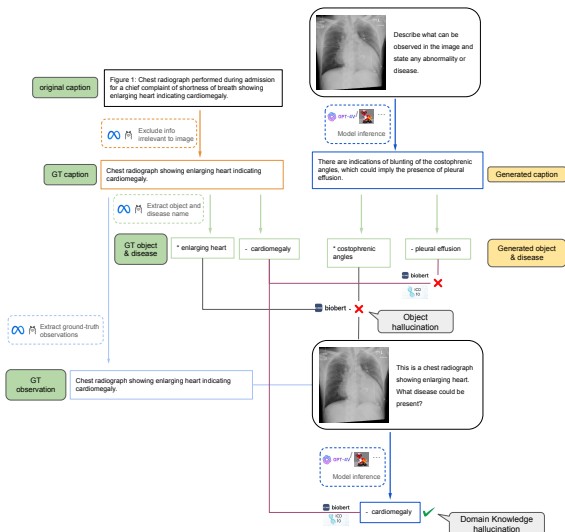

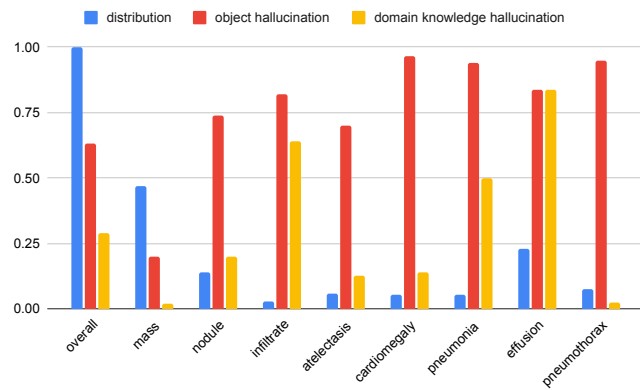

Figure 2: Distribution and hallucination of LLaVA-Med among eight chest disease categories.

Figure 1: An overview of the evaluation pipeline. The process begins with extracting a ground-truth caption from the original text, maintaining only the information pertinent to the visual content. Subsequent analysis parses this ground-truth caption to identify key objects and associated medical conditions, facilitating a comparison between model-generated captions and domain-specific knowledge.

## 2.2 Domain Knowledge Hallucination

Even when the LVLM model accurately perceives visual information, it may still produce erroneous diagnoses due to hallucinatory tendencies inherited from its pre-trained LLM component. This is akin to deducing potential diseases based on medical imaging results in a clinical context. To address this, we introduce a metric to assess hallucinations manifesting during such diagnostic reasoning processes. This metric is defined as follows:

$$\text{DKH}_I = \frac{|\ \{\ \text{hallucinated diagnosis}\ \}\ |}{|\ \{\ \text{all mentioned diagnoses}\ \}\ |} \quad (2)$$

## 3 Experiments

For preliminary analysis, we collect a subset of 1,224 image caption pairs from the MedICaT (Subramanian et al. 2020) dataset focusing on the eight chest disease categories as outlined in (Wang et al. 2017).

As shown in Fig. 1, the initial step of our evaluation pipeline involves processing the original caption using a LLM to identify the ground-truth object and disease names for each image-caption pair. Next, we engage the LVLM in a preliminary round of inference based solely on the image. We replicate the process of extracting objects and disease names from the caption generated by the LVLM and compare these with the ground-truth data. To determine the presence of object hallucination, we utilize the cosine similarity of embeddings obtained from BioBERT (Lee et al. 2019).

Our assessment combines the cosine similarity of embeddings with the distance measured using the ICD-10 coding system to establish our threshold for disease names.

If the extracted object and disease name pairs from the LVLM's first-round inference meet our set thresholds, we classify the sample as free from both object and domain knowledge hallucination. Conversely, if only the object pairs align with our threshold criteria, we categorize the sample as exhibiting solely domain knowledge hallucination.

Should the object pairs fall short of our threshold in the first round, suggesting a potential perception error, a second round of inference becomes necessary to assess the presence of domain knowledge hallucination. In this subsequent round, along with the original image, we enhance the LVLM's prompt with accurate observations extracted by the LLM from the ground-truth caption. This inclusion serves as a guide for the LVLM to make more informed diagnostic predictions based on accurately interpreted visual data in text form. If the LVLM successfully identifies the correct diagnosis in this second inference round, we consider the sample exhibiting only object hallucination.

## 4 Results

Our research shows that LLaVA-Med exhibits a 63% incidence of hallucination in object identification and 29% in domain knowledge. There appears to be no substantial correlation between these two types of hallucinations. Notably, the hallucination rate for the 'mass' category was significantly lower than for others. Upon analyzing the distribution of predicted diseases, we confirmed that the model does not disproportionately favor any single disease.

Further investigation is warranted to determine the underlying causes of these hallucinations, which could stem from visual encoding, the alignment between vision and language, or the LLM itself. One approach could involve fine-tuning the LVLM on a chest imaging dataset and comparing the hallucination rates of different model configurations. These configurations might include various combinations of vision encoders, projection layers, and LLMs before and after the fine-tuning process.

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
