# OpenReview forum: "Med-HVL: Automatic Medical Domain Hallucination Evaluation for Large Vision-Language Models"
_AAAI.org/2024/Spring_Symposium_Series/Clinical_FMs — AAAI 2024 SSS on Clinical FMs_

### Official Review · Reviewer_x5Nf · 2024-02-14
**Good pilot study focusing on caption hallucination of LVLMs. However, the manuscript needs further polish to be published.**

**Rating:** 4
**Confidence:** 5

**Review:**

The paper proposes two automatic metrics for evaluating LVLMs hallucination degree in the medical domain.

## Pros
- hallucination evaluation in the medical domain is an important research question. And the authors motivate it well.
- the proposed object hallucination and "domain knowledge hallucination" are relevant to the medical domain

## Cons
- The proposed "domain knowledge hallucination" indeed only focuses on the diagnosis. Other medical concepts such as procedures, medications, medical conditions are not included. To my understanding, it is more appropriate to use the term "diagnosis hallucination".
- The automatic evaluation is great for scaling. Meanwhile, it would be interesting to know whether these model-based metrics are really reliable (i.e. the model-derived evaluations themselves do not contain hallucination). I suggest adding some human evaluation to see (1) whether LLM-based NER is reliable; (2) whether cosine-similarity and threshold are reasonable.
- Only LLaVA-Med is evaluated, which weakens the argument presented.
- Clarification needed:
  - how to combine cosine similarity and the ICD-10-based distance?
  - On the second round inference, would the ground truth diagnosis be leaked to the LVLM in the enhanced prompt?

## Misc.
- Figure 2 is presented, but never mentioned in the text.

Overall, the proposed metrics are variants of existing metrics, with a focus on the medical domain.
The clarity can be improved to make the manuscript stronger. Certain human evaluations and LVLM evaluations are needed to thoroughly validate the technical designs.

---

### Official Review · Reviewer_iiRS · 2024-02-22

**Rating:** 6
**Confidence:** 3

**Review:**

Contribution:
The authors made two contributions in their work: 1) clear definition of hallucination within the medical field, breaking it down into two types: Object Hallucination, which involves incorrect or fabricated details about objects, and Domain Knowledge Hallucination, which pertains to inaccuracies in medical knowledge or practices. 2) they developed a new tool called Med-HVL, designed specifically for the medical domain, to detect and evaluate these hallucinations.

Pros:
Its an interesting glimpse into the bias of LLM pretrained into a large amount of medical data. A nice extension of the work would be to systematically compare and benchmark those hallucinations across multiple dataset and models.

A few remarks:
- in the figure, gt caption and gt observation are the same. Is there a gt caption without irrelevant info that would not be the gt observation? it would be nice to have a different example in the figure
- "Object" in this context seems incorrect and can be confused with support device. Maybe a more suitable term would be anatomical structures?

---

### Official Review · Reviewer_Bbeo · 2024-02-23
**This paper proposes two potentially useful metrics for evaluating medical LVLMs**

**Rating:** 7
**Confidence:** 4

**Review:**

This paper proposes two potentially useful metrics, CHAIR and DVH, for evaluating medical LVLMs. Overall this is a good metric proposal, although given the use of chest X-rays, some comparison to other metrics like RadGraph and CheXBert would be useful here.

---

### Official Review · Reviewer_FCvE · 2024-02-23
**Relevant topic, interesting approach that requires some discussion**

**Rating:** 6
**Confidence:** 4

**Review:**

In this work, the authors address a critical issue of evaluating the hallucination of LVLMs in a clinical context. As the authors assert, developing methods to assess hallucinations of these models in a quantitative and automated way is important. For this, the authors employ the CHAIR metric used in image captioning and propose a new domain knowledge hallucination metric. The authors present an initial evaluation of LLaVA-Med using the metric on the MedICAT dataset.

Comments:

- The authors address an important issue regarding hallucination of LVLMs in a clinical context.
- The proposed metric seems reasonable, however, the fact that an additional LLM is used obtain ground truth object observations is questionable. LLMs themselves are not fully validated in terms of their performance, so a proper evaluation/validation study of this step is necessary
- For assessing object hallucination and domain knowledge, the authors utilize cosine similarity of embeddings with BioBERT. Again, although the method addresses scalability, careful validation of this method for assessment is needed.
- Minor: the figure seems to be assessing GPT-4V whereas the text is evaluating LLaVA-Med

Overall, the work addresses an important issue and presents a reasonable set of metrics for assessing hallucination. Although the study is preliminary, the work will garner relevant discussion, in particular regarding the usage of existing models (such as BioBERT and GPT-4) for assessing other LVLMs.